# Effect of Tetraphenylborate on Physicochemical Properties of Bovine Serum Albumin

**DOI:** 10.3390/molecules26216565

**Published:** 2021-10-29

**Authors:** Ola Grabowska, Małgorzata M. Kogut, Krzysztof Żamojć, Sergey A. Samsonov, Joanna Makowska, Aleksandra Tesmar, Katarzyna Chmur, Dariusz Wyrzykowski, Lech Chmurzyński

**Affiliations:** Faculty of Chemistry, University of Gdańsk, Wita Stwosza 63, 80-308 Gdańsk, Poland; ola.grabowska@phdstud.ug.edu.pl (O.G.); malgorzata.kogut@phdstud.ug.edu.pl (M.M.K.); krzysztof.zamojc@ug.edu.pl (K.Ż.); sergey.samsonov@ug.edu.pl (S.A.S.); joanna.makowska@ug.edu.pl (J.M.); aleksandra.tesmar@ug.edu.pl (A.T.); k.chmur.820@studms.ug.edu.pl (K.C.); lech.chmurzynski@ug.edu.pl (L.C.)

**Keywords:** bovine serum albumin, sodium tetraphenylborate, sodium dodecyl sulfate, binding site, thermodynamic parameters

## Abstract

The binding interactions of bovine serum albumin (BSA) with tetraphenylborate ions ([B(Ph)_4_]^−^) have been investigated by a set of experimental methods (isothermal titration calorimetry, steady-state fluorescence spectroscopy, differential scanning calorimetry and circular dichroism spectroscopy) and molecular dynamics-based computational approaches. Two sets of structurally distinctive binding sites in BSA were found under the experimental conditions (10 mM cacodylate buffer, pH 7, 298.15 K). The obtained results, supported by the competitive interactions experiments of SDS with [B(Ph)_4_]^−^ for BSA, enabled us to find the potential binding sites in BSA. The first site is located in the subdomain I A of the protein and binds two [B(Ph)_4_]^−^ ions (log*K*_(ITC)1_ = 7.09 ± 0.10; Δ*G*_(ITC)1_ = −9.67 ± 0.14 kcal mol^−1^; Δ*H*_(ITC)1_ = −3.14 ± 0.12 kcal mol^−1^; TΔ*S*_(ITC)1_ = −6.53 kcal mol^−1^), whereas the second site is localized in the subdomain III A and binds five ions (log*K*_(ITC)2_ = 5.39 ± 0.06; Δ*G*_(ITC)2_ = −7.35 ± 0.09 kcal mol^−1^; Δ*H*_(ITC)2_ = 4.00 ± 0.14 kcal mol^−1^; TΔ*S*_(ITC)2_ = 11.3 kcal mol^−1^). The formation of the {[B(Ph)_4_]^−^}–BSA complex results in an increase in the thermal stability of the alfa-helical content, correlating with the saturation of the particular BSA binding sites, thus hindering its thermal unfolding.

## 1. Introduction

In aqueous media, proteins undergo many physicochemical changes such as conformational alterations, denaturation, folding/unfolding processes, and ligand exchange [1,2,3,4,5,6]. These phenomena can be invoked by variations in the temperature and/or the pH of a solution. Consequently, the environmental conditions may affect the binding properties of a biomolecule, such as the affinity of low-molecular weight compounds to a protein, the number of potential binding sites, and the stoichiometry of the resulting protein–ligand complexes [7]. As a result, the biological and pharmacological activity of the protein is often highly influenced by the presence of the different ligands in the system. Generally, the nature and the concentration of some potential ligands present in the system, which are capable to compete with other compounds (for example drugs) to a biomolecule, may affect the binding properties and functions of a protein. In the field of physicochemical studies, this phenomenon is applied to the studying of the potential binding sites of BSA by competitive displacement assays [8,9]. There are many techniques used for studying the albumin–ligand interactions. However, due to the presence of two tryptophan residues in BSA, namely Trp-134 (subdomain IA) and Trp-213 (subdomain IIB) [10], fluorescence spectroscopy is the most exploited. An excellent review on the application of direct fluorescence titrations for studying serum albumin–ligand interactions provided by Macii and Biver reveals the aspects underlying the experimental challenges and recommends new experimental design to gain some additional information on the binding type and sites of a protein [11]. It has been reported that phenylbutazone and warfarin [12] can be successfully employed as probes for determining site I (located in the hydrophobic pocket of subdomain IIA) [13] whereas diazepam and flufenamic acid [14] for site II, located in the hydrophobic cavity of subdomain IIIA.

It is also worth highlighting that albumins are one of the main objects of study in the field of bioinorganic- and metallo-drug design [15,16,17]. However, in the frame of biological studies, cell culture media used in cytotoxicity assays of new compounds usually contain relatively high amount of BSA (ca. 0.04 mM). This may result in the binding interaction of the tested compound with the cell incubation media component leading to the lowering the concentration of the free (active) form of the investigated substance.

We have focused our attention on bovine serum albumin (BSA), as it represents one of the most common model molecular systems for binding studies [18]. Serum albumins are involved, among others, in transport and biodistribution of endogenous ligands such as fatty acids [19] as well as endogenous substances, for example ibuprofen, warfarin, penicillin or diazepam [11]. They also participate in the metabolism of low-molecular weight compounds. Recently, we have proven that the isothermal titration calorimetry technique supported by experimental and in silico methods can be used as an alternative to fluorescence spectroscopy for studying a stoichiometry of the resulting albumin–ligand complexes, the type of the binding interactions, and the number of the binding sites [20]. The studies performed on BSA and sodium dodecyl sulfate (SDS) have revealed that the investigated protein possesses two main binding sites: the first site is pH and temperature independent, and binds two moles of SDS and is located close to Trp-134 residue. In contrast, the total number of SDS bound to the second site of the protein depends on the pH of the solution (pH 5 and pH 7) and the temperature (290 K and 300 K) (Figure 1) [20].

In this work, we have exploited the previously gathered information on the SDS–BSA interactions for studying the binding interactions of tetraphenylborate ions ([B(Ph)_4_]^−^) with BSA. We have focused our attention on the [B(Ph)_4_]^−^ ions, as they represent the low-molecular weight compounds with four, bulky hydrophobic phenyl moieties and a negative charge capable of binding through hydrophobic interactions and/or a combination of hydrophobic and electrostatic forces [21] (Figure 2). A set of some complementary methods, namely Isothermal Titration Calorimetry (ITC), steady-state fluorescence spectroscopy, Differential Scanning Calorimetry (DSC) and Circular Dichroism (CD) was employed for assessing the physicochemical nature of the interactions. The effect of the [B(Ph)_4_]^−^ ions binding on the protein structure was also discussed. Finally, new competition ITC experiments using SDS as a competitive ligand have been performed to find BSA binding sites of [B(Ph)_4_]^−^ ions. Then, the experimental results were subsequently verified by a Molecular Dynamics (MD) study. The obtained results of the [B(Ph)_4_]^−^–BSA interactions were discussed in relation to a previous published study on the interaction of sodium dodecyl sulfate (SDS) with BSA.

## 2. Results and Discussion

### 2.1. Isothermal Titration Calorimetry (ITC)

Thermodynamic parameters of the BSA interaction with [B(Ph)_4_]^−^ ions determined by the ITC technique in the cacodylate buffer solution of a pH 7, at 298.15 K, were subsequently compared with the results obtained under the same experimental conditions for the BSA–SDS system [20]. In both investigated cases, namely {[B(Ph)_4_]^−^}–BSA and SDS-BSA, nonlinear least-squares procedures were applied for fitting isotherms to a model that assumes two sets of binding sites. The stoichiometry of the ligand–albumin complexes (*N = [ligand]/[BSA]*), binding constants (*K*_ITC_) and the enthalpy change (Δ*H*_ITC_) were obtained directly from ITC measurements. The free energy of binding (Δ*G*_ITC_) and entropy change (Δ*S*_ITC_) were calculated using the standard thermodynamic relationships: Δ*G*_ITC_ = −R*T*ln*K*_ITC_ = Δ*H*_ITC_-TΔ*S*_ITC_. The obtained parameters are so-called condition-dependent parameters and can be compared with those obtained under the same pH, temperature and the type of a buffer solution [22]. Representative binding isotherms for the investigated systems are shown in Figure 3, whereas conditional parameters of the interactions are summarized in Table 1. 

ITC data revealed that BSA has two binding sites capable of binding tetraphenylborate ions, as has been previously observed for SDS monomers (Figure 3, Table 1). The simplified chemical equations of the ligand (SDS or [B(Ph)_4_]^−^)–BSA complex formation which take into account the stoichiometry of the resulting species are given below:

The first binding site
2 SDS + BSA = (SDS)_2_-BSA           log*K*_(ITC)1_ = 7.61
2 Na[B(Ph)_4_] + BSA = (Na[B(Ph)_4_])_2_-BSA     log*K*_(ITC)1_ = 7.09

The second binding site
5 SDS + (SDS)_2_-BSA = (SDS)_2_-BSA-(SDS)_5_     log*K*_(ITC)2_ = 5.29
4 Na[B(Ph)_4_] + (Na[B(Ph)_4_])_2_-BSA = (Na[B(Ph)_4_])_2_-BSA-(Na[B(Ph)_4_])_4_  log*K*_(ITC)2_ = 5.39

The first site of albumin binds two [B(Ph)_4_]^−^ ions followed by the saturation of the second site by four tetraphenylborate ions. The binding constants (*K*_(ITC)_), and consequently the free energy of binding (Δ*G*_ITC_), for both SDS–BSA and [B(Ph)_4_]^−^–BSA complexes are comparable to one another (in the range of experimental error) indicating their similar thermodynamic stability (Table 1). In contrast to the SDS–BSA interactions in which the charge–charge type or/and van der Waals interactions are involved in the ligand–protein complex formation (|Δ*H*_(ITC)_| > |*T*Δ*S*_(ITC)_|), the binding of [B(Ph)_4_]^−^ ions in the first site of the albumin is both enthalpy- and entropy-driven. However, the hydrophobic interactions seem to prevail (|Δ*H*_(ITC)_| < |*T*Δ*S*_(ITC)_|) [23].

A different scenario is observed for binding [B(Ph)_4_]^−^ ions in the second site. The positive value of Δ*H*_(ITC)2_ shows that the thermodynamic stability of the resulting complexes strongly depends on the entropy change, while the charge–charge type or/and van der Waals interactions are involved in the SDS binding to the second site (Table 1). Therefore, it is supposed that [B(Ph)_4_]^−^-BSA (2nd site) binding occurs mainly via hydrophobic interactions which contribute to an increase in entropy on account of the dehydration of the reactants or as a result of ions exchange occurring in the [B(Ph)_4_]^−^ binding process. Subsequently, this supposition has been verified by MD simulations.

The obtained findings were employed for studying the potential [B(Ph)_4_]^−^ binding sites on the surface of the protein. To do this, the first and then the second binding site were saturated with [B(Ph)_4_]^−^ ions, and checked for whether the albumin is still able to bind SDS monomers (Figure 4). It turned out that BSA with the first site saturated by [B(Ph)_4_]^−^ ions is still able to bind SDS (log*K*_(ITC)_ = 4.92 ± 0.04; Δ*G*_(ITC)_ = −6.73 ± 0.05 kcal mol^−1^; Δ*H*_(ITC)_ = −12.85 ± 0.35 kcal mol^−1^; *T*Δ*S*_(ITC)_ = −6.12 kcal mol^−1^), but in such a system only around four to five molecules of SDS are bound to one site, as opposed to around seven (*N*_1_ ~ 2, *N*_2_ ~ 5), as was observed in the absence of [B(Ph)_4_]^−^ ions in the solution (Table 1).

Based on these observations, it can be assumed that both ligands ([B(Ph)_4_]^−^ and SDS) reveal an affinity to the same “first” binding site in BSA. After a saturation of the first site by [B(Ph)_4_]^−^ there is not enough space for binding of SDS monomers to the same surface of BSA. For this reason, negatively charged SDS anions are forced to bind to the second site of the protein. The above assumption confirms the following observations:(a)There are no significant differences in the thermodynamic stability of the ligand–albumin complexes formed in the first binding site. Thus, SDS is not a strong enough ligand to replace [B(Ph)_4_]^−^ in the first binding site.(b)Binding constants of the SDS–BSA complexes formed in the absence and in the presence of [B(Ph)_4_]^−^ have similar values:
5 SDS + (SDS)_2_-BSA = (SDS)_2_-BSA-(SDS)_5_           log*K*_(ITC)2_ = 5.29 (±0.06)
5 SDS + (Na[B(Ph)_4_])_2_-BSA = (Na[B(Ph)_4_])_2_-BSA-(SDS)_5_     log*K*_(ITC)_ = 4.92 (±0.04)In the absence of [B(Ph)_4_]^−^ in the system, the binding constant of SDS to the first binding site of BSA is expected to be log*K*_(ITC)_ approximately 7 (Table 1). The ITC data showed that in cases where the [B(Ph)_4_]^−^ ions occupy the first site of the protein, the calculated binding constant (log*K*_(ITC)_ ~ 5) corresponds to the binding of SDS to the second binding site.(c)The repulsion interactions between the negatively charged ligands ([B(Ph)4]^−^ and SDS) disfavor occupation by them the same binding site in BSA.

A similar phenomenon has been observed after the saturation of the albumin by [B(Ph)_4_]^−^ in both binding sites (Figure 4). Under such experimental conditions, the albumin still possesses one binding site capable of binding four moles of SDS monomers (log*K*_(ITC)_ = 4.97 ± 0.07; Δ*G*_(ITC)_ = −6.78 ± 0.09 kcal mol^−1^; Δ*H*_(ITC)_ = −13.95 ± 0.71 kcal mol^−1^; TΔ*S*_(ITC)_ = −7.17 kcal mol^−1^), according to the following general equation:4 SDS + {(Na[B(Ph)_4_])_2_-(Na[B(Ph)_4_])_4_}(BSA) = {(Na[B(Ph)_4_])_2_-(Na[B(Ph)_4_])_4_}(BSA)(SDS)_4_

Moreover, it is worth noticing that the parameters of the interactions between SDS and BSA within the first as well as both sites being saturated by [B(Ph)_4_]^−^ are comparable (within the range of the experimental error). This confirms the fact that the investigated ligands show affinity only for the first site in BSA. Upon saturation of the same “first” site in BSA, [B(Ph)_4_]^−^ and SDS are bound at different sites.

### 2.2. Steady-State Fluorescence Spectroscopy

Figure 5 presents the fluorescence emission spectra of free BSA and the protein mixed with Na[B(Ph)_4_] in molar ratios equal to 1:3, 1:7, and 1:15 under the action of increasing concentrations of SDS. First of all, an increase in the amount of [B(Ph)_4_]^−^ in the system has no impact on the position of an initial strong emission band of tryptophan residues in BSA at approximately 348 nm. Secondly, it can be observed that the intrinsic fluorescence intensity of the albumin (free and saturated with [B(Ph)_4_]^−^) decreases regularly with the addition of SDS, accompanied by a significant blue shift from 348 nm to 332 nm, which may be a result of conformational changes in the BSA structure under the action of the surfactant (exposure of tryptophan residues to a more hydrophobic environment) [24]. Furthermore, the higher the amount of [B(Ph)_4_]^−^ ions in the mixture, the lower the changes in the fluorescence intensity of BSA under the action of SDS. These findings are in line with the ITC results, and confirm the fact that upon saturation of BSA with [B(Ph)_4_]^−^ the changes in the Trp fluorescence intensity are not as pronounced as for the [B(Ph)_4_]^−^-free solution. This is related to the fact that under such experimental conditions, the total number of SDS monomers bound to the protein is lower, and the SDS monomers are bound to their second binding site, which is far from sensitive to SDS binding of the Trp-134 and Trp-213 residues [20].

To obtain a better insight into the mechanisms of the interactions between BSA (and its mixtures with [B(Ph)_4_]^−^) and SDS, the obtained results were analyzed according to a well-known Stern–Volmer equation: $\frac{F_{0}}{F}=1+K_{SV}\left[Q\right]=1+k_{q}\tau_{0}\left[Q\right]$, where F_0_ and F denote the fluorescence intensities in the absence and presence of a quencher (SDS), [Q] is the quencher concentration, *K*_SV_ is the Stern–Volmer quenching constant, k_q_ is the bimolecular quenching rate constant, while τ_0_ is the lifetime of the fluorophore (BSA) in the absence of quencher [25,26]. The graphs of $\frac{F_{0}}{F}$ versus [Q] plotted according to the Stern–Volmer equation are shown in Figure 6. Consequently, Table 2 presents the newly determined values of Stern–Volmer quenching constants along with linear correlation coefficients (R^2^) and bimolecular quenching rate constants (k_q_) recovered for the studied systems. The latter ones were calculated based on the value of the average fluorescence lifetime τ_0_ of free BSA (in Tris-HCl, pH 7.0), which equals approximately 6.3 ns [27]. It can be observed that the quenching (binding) constants decrease with the increase in the [B(Ph)_4_]^−^ concentration of the system, which can be assigned to the competition of SDS with [B(Ph)_4_]^−^ in BSA [28,29]. Furthermore, the estimated values of k_q_ for all systems are of the order 10^12^ M^−1^·s^−1^, which is approximately a few hundred times higher than the maximum value possible for the diffusion-controlled quenching-rate constant (2.0 × 10^10^ M^−1^·s^−1^) [30]. Since the values of bimolecular quenching rate constants are considered to be definitive in differentiating between dynamic and static quenching mechanisms, the predominant role of ground-state complexation (static quenching) in the investigated systems was confirmum albumin was measured using an extinction coefficient eqed.

### 2.3. Differential Scanning Calorimetry (DSC)

The influence of [B(Ph)_4_]^−^ ions on a thermal unfolding transition of BSA was assessed based on DSC experiments. The representative, original DSC curves are shown in Figure 7. Two distinct peaks are observed in the heat-capacity curve for the sample of free BSA. The transition midpoints *T*_m1_ = 328.9 K and *T*_m2_ = 346.7 K were estimated by fitting the experimental C_p_ vs. *T* curves to a non-two-state model using Marquardt nonlinear least-squares method (*T*_m_ denotes the temperature value at which a maximum endothermic effect appears). Previously, it has been reported that the appearance of these peaks corresponds to the low-temperature (*T*_m1_) and the high-temperature (*T*_m2_) transition resulting from the melting of structurally independent parts of the protein [31,32,33,34]. The less thermally stable domain comprises the subdomains IIB, IIIA, and IIIB, whereas the subdomains IA, IB, and IIA represent the region (the energetic domain) more resistant to thermal unfolding [35].

The formation of fairly stable Na[B(Ph)_4_]–BSA complexes makes the BSA structure more compact and thermostable. This is reflected in the shift of the heat-induced transitions towards a higher range of temperatures with the increase in the Na[B(Ph)_4_]:BSA molar ratios in the mixture, namely *T*_m1_ = 341.8 K and *T*_m2_ = 351.9 K for the saturation of the first binding sites (Na[B(Ph)_4_]:BSA = 2.5:1) and *T*_m1_ = 349.2 K and *T*_m2_ = 356.2 K for BSA with both binding sites saturated (Na[B(Ph)_4_]:BSA = 7:1).

### 2.4. Circular Dichroism (CD) Spectra Analysis

The effect of Na[B(Ph)_4_] ions’ binding and temperature on the secondary structure of BSA was assessed by temperature-dependent CD measurements in the range of temperatures from 298.15 K to 368.15 K (Figure 8, Figure 9 and Figure 10). The performed experiments showed that the saturation of BSA with [B(Ph)_4_]^−^ ions at 298.15 K does not contribute to noticeable changes in the secondary structure of the protein. On the other hand, the thermal stability of the α-helix structure increases on account of the formation of the Na[B(Ph)_4_]–BSA complexes (Table 3). These findings are in agreement with the DSC results.

### 2.5. Molecular Modelling

To gain a deeper understanding of the experimental results at the atomistic level, MD simulation was conducted for the [B(Ph)_4_]^−^–BSA system. In particular, the MD approach was applied to predict potential binding sites of [B(Ph)_4_]^−^ on the BSA surface and to further characterize and compare them in terms of the binding affinities for these ions. The protein was surrounded by 15 randomly distributed [B(Ph)_4_]^−^ ions and the simulation was performed for 100 ns at 300 K and a pH of 7. After the careful examination of the frames for each [B(Ph)_4_]^−^ ligand, the last 40% of the simulation was used for the further LIE analysis (Table 4). Based on our results, two bindings sites were localized for three molecules (Figure 11). Interestingly, one binding site is in the proximity to Trp-134, as was observed in the case of our previous SDS–BSA study [20]. Two ions bound to site I and one ion to site II after 40 and 60 ns, respectively, and remained there until the end of the simulation. Their RMSD as well as the RMSD of the protein are shown in Appendix A to demonstrate the convergence. LIE energy values for the ions bound in sites I and II are shown in Appendix A, supporting the convergence achieved for the binding in these sites. In order to strictly define the binding event in sites I and II, a 5 Å cutoff criteria was applied for the distance between the residues of the sites and [B(Ph)_4_]^−^ ion atoms. According to LIE calculations, site I has slightly higher affinity towards [B(Ph)_4_]^−^ than site II due to the more favorable van der Waals component of the free binding energy (the electrostatic and van der Waals components calculated for each [B(Ph)_4_]^−^ ion are provided in Table 4). Site I comprises Lys116, Pro119, Glu140, Ile141, Arg144, His145, Leu178, Glu182, Arg185, and Val188, while site II is formed by Leu386, Asn390, Phe402, Leu406, Arg409, Thr410, Lys413, Thr448, Ser479, Arg484, and Phe487. Clearly, site I has a more charged nature and lower propensity for establishing pi–pi and cation–pi interactions than site II. Additionally, when looking at the right panel of Figure 11, it seems that theoretical calculations confirm that hydrophobic rings of phenyl groups contribute to the stabilization of interactions between BSA and [B(Ph)_4_]^−^.

## 3. Materials and Methods

### 3.1. Reagents

Bovine serum albumin (BSA, lyophilized powder, ≥96%), sodium dodecyl sulfate (SDS, for molecular biology, ≥99%), sodium tetraphenylborate (Na[B(Ph)_4_], ≥99.5%), sodium dodecyl sulfate (SDS, for molecular biology, ≥99%) and sodium cacodylate trihydrate (Caco, ≥98%) were obtained from Merck (Poland) and employed as received without further purification. Double-distilled water with conductivity not exceeding 0.18 μS cm^−1^ was used for preparations of buffer solutions.

### 3.2. Isothermal Titration Calorimetry (ITC)

All ITC experiments were performed at 298.15 K using an AutoITC isothermal titration calorimeter (MicroCal Inc. GE Healthcare, Northampton, MA, USA). The details of the measuring device and experimental setup have been described previously [20]. The reagents were dissolved directly in 10 mM Caco buffer of pH 7. The experiments consisted of injecting 10.02 μL (29 injections, 2 μL for the first injection only) of (a) 1 mM buffered solution of Na[B(Ph)_4_] into the reaction cell which initially contained the 0.02 mM buffered solution of BSA, and (b) 1 mM buffered solution of SDS into the reaction cell which initially contained the Na[B(Ph)_4_]:BSA mixture of the 3:1 (0.0375 mM Na[B(Ph)_4_], 0.0125 mM BSA) or 7:1 (0.0875 mM Na[B(Ph)_4_], 0.0125 mM BSA) molar ratio. A background titration, consisting of an identical titrant solution but with the buffer solution in the reaction cell only, was subtracted from each experimental titration on account of the heat of dilution. The titrant was injected at 5 min intervals. Each injection lasted 20 s.

### 3.3. Steady-State Fluorescence Spectroscopy and UV Spectrophotometry

The stock solutions of BSA, SDS and Na[B(Ph)_4_] were prepared in 10 mM Caco buffer of pH 7 (all subsequent dilutions were made with this buffer). The concentration of bovine serum albumin was measured using an extinction coefficient equal to $\varepsilon_{280}^{BSA}=41,180\;M^{-1}{\mathrm{cm}}^{-1}$, calculated based on the content of tryptophan ($\varepsilon_{280}^{W}=5690\;M^{-1}{\mathrm{cm}}^{-1}$), tyrosine ($\varepsilon_{280}^{Y}=1280\;M^{-1}{\mathrm{cm}}^{-1}$), and cysteine ($\varepsilon_{280}^{C}=120\;M^{-1}{\mathrm{cm}}^{-1}$) [36]. A maximum absorbance value of approximately 0.08 at 280 nm (corresponding to a protein concentration of 2 µM) was used to avoid the inner-filter effect.

Steady-state fluorescence experiments were carried out with a Cary Eclipse Varian (Agilent, Santa Clara, CA, USA) spectrofluorometer, equipped with a temperature controller and a 1.0 cm multicell holder. The absorption spectra were recorded on Perkin Elmer Lambda 650 (Waltham, MA, USA) UV/Vis spectrophotometer. In the performed fluorescence titration experiments: (i) 2 mL of pure BSA (2 µM BSA); (ii) 2 mL of Na[B(Ph)_4_] and BSA mixed in ratio 3:1 (6 µM Na[B(Ph)_4_], 2 µM BSA); (iii) 2 mL of BSA and Na[B(Ph)_4_] mixed in ratio 7:1 (14 µM Na[B(Ph)_4_], 2 µM BSA); and (iv) 2 mL of BSA and Na[B(Ph)_4_] mixed in ratio 15:1 (30 µM Na[B(Ph)_4_], 2 µM BSA) were simultaneously titrated with ten 5 μL aliquots of sodium dodecyl sulphate (1 mM). The fluorescence intensity of the band at 348 nm—corresponding to the initial maximum emission of BSA—was used to calculate the quenching constants and then other parameters. Excitation wavelength was always set at 280 nm. All experiments were performed at 298.15 K.

### 3.4. Differential Scanning Calorimetry (DSC)

Calorimetric (DSC) measurements were made with a VP-DSC microcalorimeter (MicroCal Inc. GE Healthcare, Northampton, USA) at a scanning rate 90 K h^−1^. All scans were run in 10 mM Caco buffer of pH 7 in the temperature range of 298.15–363.15 K at a scan rate of 1.5 K min^−1^. The scans were obtained for pure BSA (0.015 mM) and for the solution containing Na[B(Ph)_4_] and BSA mixed in ratio 2.5:1 (0.0375 mM Na[B(Ph)_4_], 0.015 mM BSA) and 7:1 (0.105 mM Na[B(Ph)_4_], 0.015 mM BSA). All the samples were degassed before measurements. These measurements were recorded two times. The details of the measuring device, experimental setup and data processing have been described previously [20].

### 3.5. Circular Dichroism Spectroscopy (CD)

Circular dichroism (CD) spectra were recorded in water on a Jasco-715 automatic recording spectropolarimeter (Jasco Inc., Easton, MD, USA) for the following systems: pure BSA (0.0015 mM) and two Na[B(Ph)_4_]/BSA mixtures in stoichiometric molar ratios 2.5:1 (0.00375 mM Na[B(Ph)_4_], 0.0015 mM BSA) and 7:1 (0.0105 mM Na[B(Ph)_4_], 0.0015 mM BSA) in the 10 mM Caco buffer of pH 7, in a temperature range from 298.15 K to 368.15 K, every 10 degrees. The details of the measuring device, experimental setup and data processing have been described previously [20]. The secondary structure of BSA under experimental conditions in the absence and presence of Na[B(Ph)_4_] was determined using the CONTIN/LL program, a variant of the CONTIN method developed by Provencher and Glockner [37] provided within the CDPro software package.

### 3.6. Theoretical Studies

#### 3.6.1. Structures

The structure of Bovine Serum Albumin (BSA) was obtained from the Protein Data Bank (PDB ID: 4F5S, 2.47 Å). The structure of [B(Ph)_4_]^−^ (referred to as BPH for simplicity) was parameterized as by Kurt and Temel [38,39] and later built in xleap of AMBER16 package [40].

#### 3.6.2. Molecular Dynamics Simulations

The MD approach was applied to localize potential binding sites and to characterize them in terms of the binding affinities. There are several reasons why such an approach is superior to the classical molecular docking approach, and therefore was used instead of molecular docking in our study: (1) Docking small ligands as SDS in our previous work [20] and [B(Ph)_4_]^−^ to BSA in this work did not allow for finding more than a single binding site. (2) MD allows for the complete flexibility of the protein residues, which could be crucial for the binding of a small ligand, while rigid receptor molecular docking does not allow for it. Taking into account a structural ensemble of receptors in molecular docking is a possible alternative which is computationally more expensive than the MD approach. (3) In contrast to molecular docking, solvent is treated explicitly in the MD simulations. (4) Molecular docking scores are far less reliable than MD-derived binding affinities due to the fact that molecular docking scores single, static poses, while multiple frames corresponding to the statistical ensemble of the structures are considered in the MD-based calculations. (5) In most of the state-of-the-art modelling studies, the poses obtained by a molecular docking approach are further analyzed with the MD approach. Therefore, the MD approach, when applied for predicting binding sites, is already self-consistent in comparison to the two-step procedure that includes molecular docking followed by MD. Hence, the MD approach is preferable. (6) Furthermore, we previously demonstrated both convergence and high predictive power of the MD approach when applied to the protein–ion systems [41]. All all-atom molecular dynamics (MD) simulations were performed with the use of AMBER16 software package [40]. The initial structures of the complexes were created by random placement of 15 [B(Ph)_4_]^−^ ions around the BSA protein. A truncated, octahedron, TIP3P periodic box of an 8 Å water layer from the box’s border to the solute was used to solvate complexes. The ff14SBonlysc force field for the protein [42] and gaff force field [43] with RESP charges [44] obtained in the antechamber module of AMBER16 [40] for [B(Ph)_4_]^−^ were used. The net negative charge of the system was neutralized with Na^+^ counterions. Energy minimization was carried out in two steps: beginning with 500 steepest descent cycles and 10^3^ conjugate gradient cycles with 100 kcal/mol/Å^2^ harmonic force restraints on solute atoms, continued with 3 × 10^3^ steepest descent cycles and 3 × 10^3^ conjugate gradient cycles without any restraints. Following minimization steps, the system was heated up from 0 to 300 K for 10 ps with harmonic force restraints of 100 kcal/mol/Å^2^ on solute atoms. Then, the system was equilibrated at 300 K and 10^5^ Pa in isothermal isobaric ensemble for 100 ps. Afterwards, an independent prediction MD run was carried out in the same isothermal isobaric ensemble for 100 ns. The particle mesh Ewald method for treating electrostatics and the SHAKE algorithm for all the covalent bonds containing hydrogen atoms were implemented in the MD simulations. Such parameters within the MD simulation were shown to be sufficient for a system of ions with a protein, which was substantially bigger than BSA, allowing for proper prediction of the ion binding sites [41].

#### 3.6.3. Binding Free-Energy Calculations

Energetic postprocessing of the trajectories and per-residue energy decomposition was performed for the system with the use of Linear Interaction Energy (LIE) with dielectric constant of 80. Frames from the MD simulation were carefully examined with VMD [45]. The last 40% of the simulation was chosen for the analysis, since in the corresponding frames no more events of ion dissociations or associations were observed. LIE takes into account only the electrostatic component (scaled by the dielectric constant) and the van der Waals component of the binding free energy. Although the binding free energies obtained by this method are not expected to represent absolute binding free energies, these calculations are useful to compare the binding strength in different binding sites.

## 4. Conclusions

A few complementary experimental methods supported by in silico analysis have successfully been applied to describe the physicochemical nature of tetraphenylborate ions’ interactions with BSA. The obtained results were subsequently discussed in relation to a previously studied system, namely the interaction of sodium dodecyl sulphate (SDS) with BSA. The described approach enables us to find the BSA potential binding sites of [B(Ph)_4_]^−^.

In conclusion, two main binding sites of [B(Ph)_4_]^−^ were unveiled. They are located in the subdomain IA and IIIA, respectively, of the investigated protein. Both tetraphenylborate ions and SDS molecules reveal a similar affinity to the first specific binding site in BSA which is located close to the Trp-134 residue. In turn, the second binding site on the BSA surface for investigated ligands are different. The binding of [B(Ph)_4_]^−^ ions in the subdomain IIIA is an entropy-driven process, whereas SDS in the subdomain IIB is enthalpy-driven.

It has been proven that [B(Ph)_4_]^−^ ions participate in the stabilization of the tertiary structure of BSA at higher temperatures on account of the formation of fairly stable [B(Ph)_4_]^−^–BSA complexes of a different stoichiometry ([B(Ph)_4_]^−^:BSA = 2:1 and 5:1). This phenomenon underlying its biological activity could be responsible for the protection of the protein from thermal unfolding.

The results clearly show that the presence of competitive ligands may have some implications on the concentration of the compounds tested (as has been found for [B(Ph)_4_]^−^ ions or SDS) in the cell-culture media containing albumin. On the one hand, BSA, an important component of the cell culture media, binding to the compounds of a hydrophobic nature, as [B(Ph)_4_]^−^ or SDS, can lead to a decrease in the concentration of free, active species. In consequence, this can affect the cytotoxic action of the tested compound. On the other hand, the saturation of the potential binding sites of BSA by a nontoxic, low-molecular-weight ligand may prevent the interactions of albumin with other biologically active compounds, thus affecting their dose-dependent action.

The findings presented in this contribution are worth taking into consideration during the analysis of the cytotoxicity assays of the compounds being characterized.

## Figures and Tables

**Figure 1 molecules-26-06565-f001:**
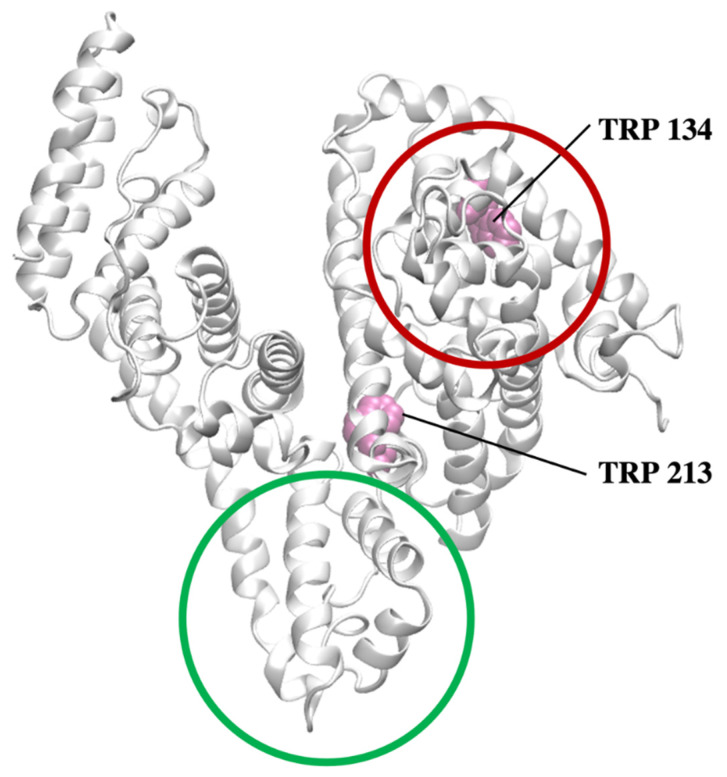
The secondary structure of BSA (grey cartoon) with marked Trp residues (magenta VDW representation). The first and second potential binding sites of SDS are shown with red and green rings, respectively.

**Figure 2 molecules-26-06565-f002:**
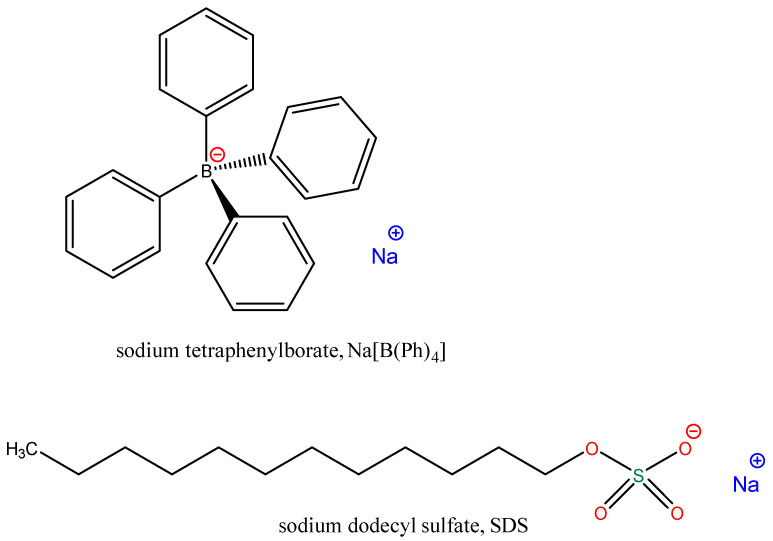
Chemical structure of sodium tetraphenylborate (Na[B(Ph)_4_]) and sodium dodecyl sulfate (SDS).

**Figure 3 molecules-26-06565-f003:**
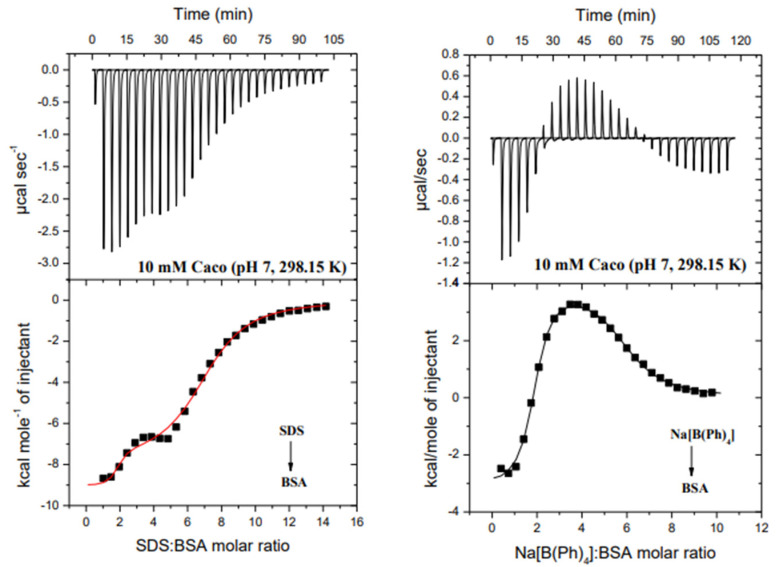
Calorimetric titration isotherms of the binding interactions between SDS and BSA (LEFT) and Na[B(Ph)_4_] and BSA (RIGHT) in the10 mM Caco buffer of pH 7, at 298.15 K.

**Figure 4 molecules-26-06565-f004:**
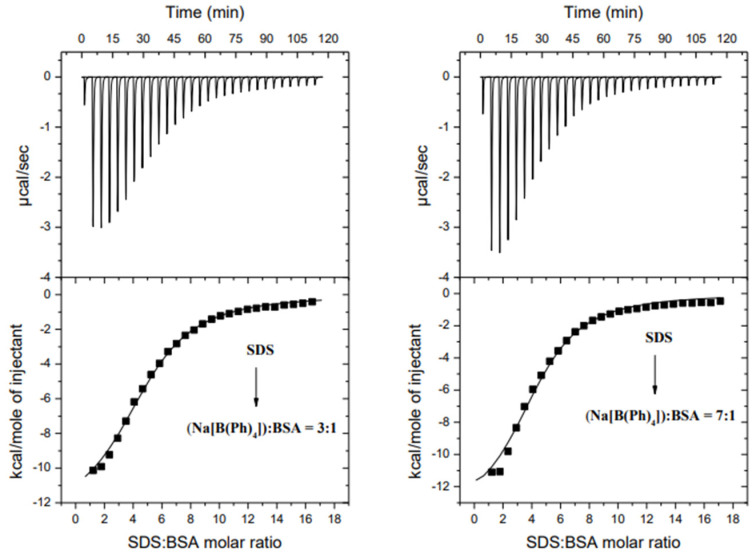
Calorimetric titration isotherms of the binding interactions between SDS and BSA in the presence of Na[B(Ph)_4_] in the Na[B(Ph)_4_]:BSA molar ratio 3:1 (**left**) and 7:1 (**right**) in the 10 mM Caco buffer of pH 7, at 298.15 K.

**Figure 5 molecules-26-06565-f005:**
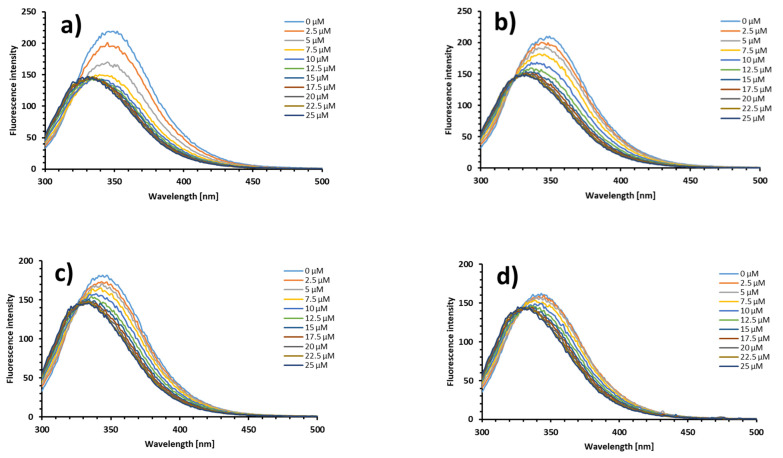
The fluorescence emission spectra of free BSA (**a**) and the solutions of BSA with Na[B(Ph)_4_] mixed in molar ratios 1:3 (**b**); 1:7 (**c**); and 1:15 (**d**) in the presence of increasing concentrations of SDS (0–25 μM) in the 10 mM Caco buffer of pH 7.0 at 298.15 K.

**Figure 6 molecules-26-06565-f006:**
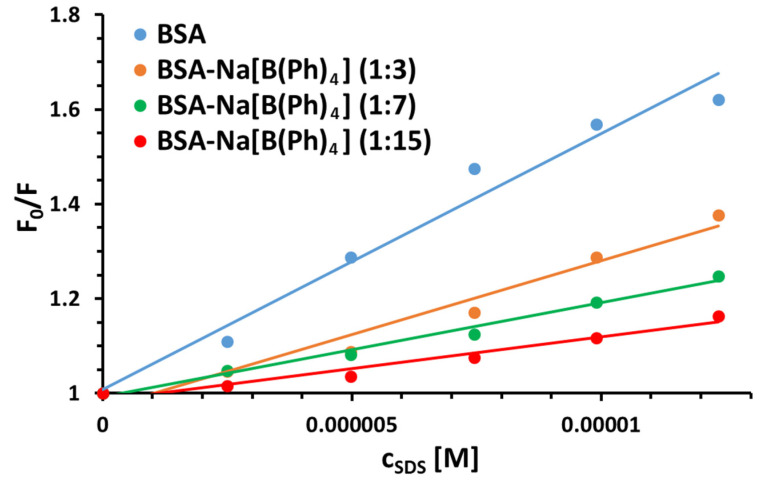
Stern–Volmer plots for the steady-state fluorescence quenching of BSA and its mixtures with Na[B(Ph)_4_] (1:3; 1:7; and 1:15) by SDS in the 10 mM Caco buffer of pH 7.0 at 298 K.

**Figure 7 molecules-26-06565-f007:**
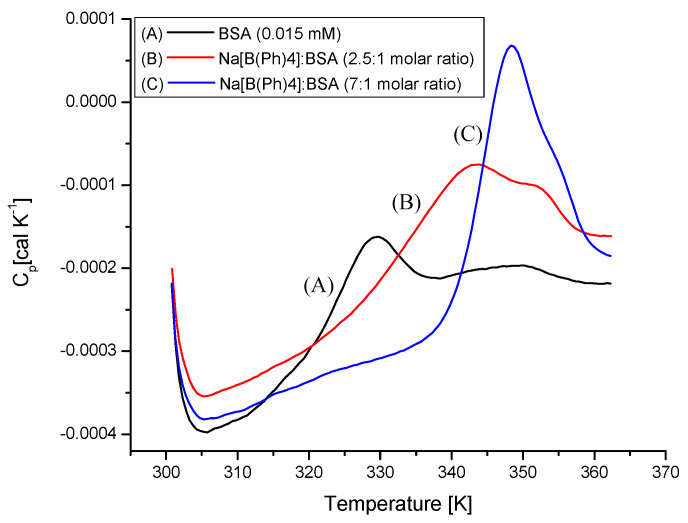
The raw heat capacity data for: (A) free BSA (0.015 mM) and the mixtures (B) with the Na[B(Ph)_4_]:BSA molar ratios 2.5:1 (0.0375 mM Na[B(Ph)_4_], 0.015 mM BSA) and (C) 7:1 (0.105 mM Na[B(Ph)_4_], 0.015 mM BSA) in the 10 mM Caco buffer of pH 7.

**Figure 8 molecules-26-06565-f008:**
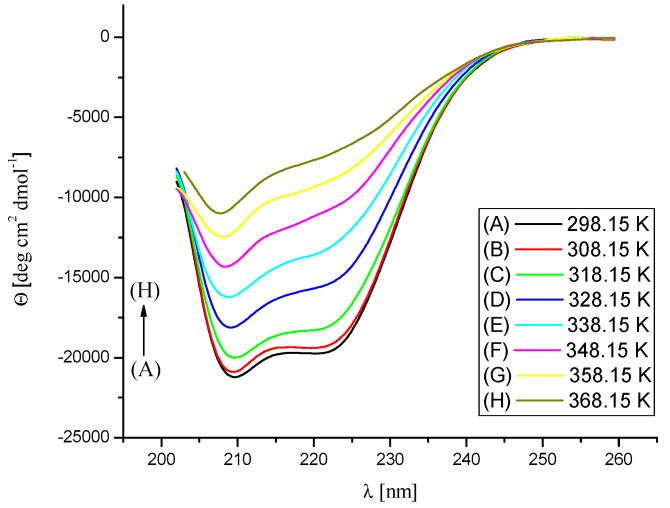
CD spectra of BSA in 10 mM Caco buffer at pH 7, in the temperature range 298.15–368.15 K. The concentration of BSA was maintained at 0.0015 mM.

**Figure 9 molecules-26-06565-f009:**
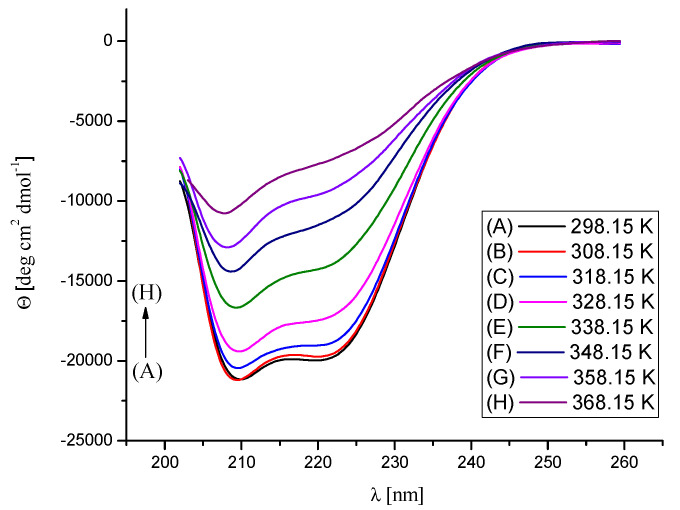
CD spectra of the Na[B(Ph)_4_]/BSA mixture (Na[B(Ph)_4_]:BSA = 2.5:1 molar ratio) in 10 mM Caco buffer at pH 7, in the temperature range 298.15–368.15 K. The concentration of BSA was maintained at 0.0015 mM.

**Figure 10 molecules-26-06565-f010:**
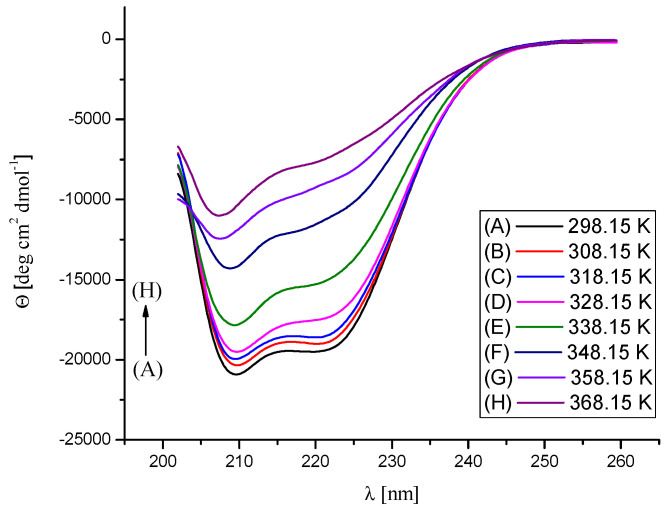
CD spectra of the Na[B(Ph)_4_]/BSA mixture (Na[B(Ph)_4_]:BSA = 7:1 molar ratio) in 10 mM Caco buffer at pH 7, in the temperature range 298.15–368.15 K. The concentration of BSA was maintained at 0.0015 mM.

**Figure 11 molecules-26-06565-f011:**
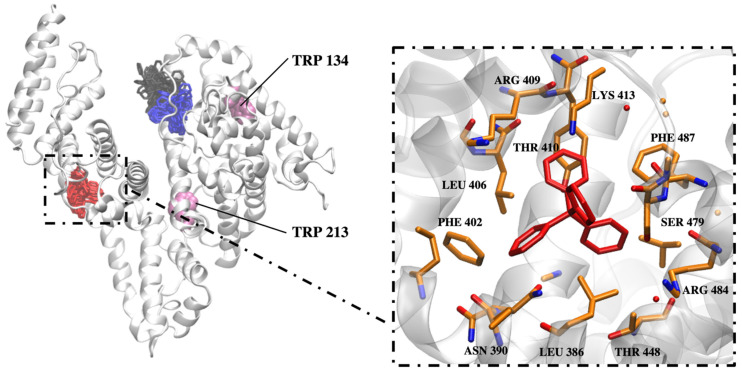
On the left, BSA protein (grey cartoon), Trp residues (magenta VDW representation) and three clusters (licorice) representing [B(Ph)_4_]^−^ ions that bound to the protein with the highest affinity. Colors of the clusters correspond to the color coding used in Table 4. On the right panel, a representative frame for one [B(Ph)_4_]^−^ ion and its protein surrounding.

**Table 1 molecules-26-06565-t001:** The conditional thermodynamic parameters of SDS and Na[B(Ph)_4_] binding to BSA (standard deviation values in parentheses) in the 10 mM Caco buffer of pH 7, at 298.15 K.

Parameter	SDS/BSA ^(1)^	Na[B(Ph)_4_]/BSA
*N* _1_	1.75 (±0.14)	1.78 (±0.02)
log*K*_(ITC)1_	7.61 (±0.38)	7.09 (±0.10)
Δ*G*_(ITC)1_ [kcal mol^−1^]	−10.38 (±0.67)	−9.67 (±0.14)
Δ*H*_(ITC)1_ [kcal mol^−1^]	−9.11 (±0.26)	−3.14 (±0.12)
TΔ*S*_(ITC)1_ [kcal mol^−1^]	1.27	6.53
*N* _2_	5.33 (±0.29)	4.09 (±0.09)
log*K*_(ITC)2_	5.29 (±0.06)	5.39 (±0.06)
Δ*G*_(ITC)2_ [kcal mol^−1^]	−7.22 (±0.09)	−7.35 (±0.09)
Δ*H*_(ITC)2_ [kcal mol^−1^]	−7.98 (±0.27)	4.00 (±0.14)
TΔ*S*_(ITC)2_ [kcal mol^−1^]	−0.76	11.3

^(1)^ Literature data [17].

**Table 2 molecules-26-06565-t002:** Stern–Volmer quenching constants, linear correlation coefficients (R^2^) and bimolecular quenching rate constants (k_q_) recovered for the steady-state fluorescence quenching of pure BSA and its mixtures with Na[B(Ph)_4_] (1:3; 1:7; and 1:15) by SDS in the 10 mM Caco buffer of pH 7.0 at 298 K.

System	*K*_SV_ [10^4^ M^−1^]	R_2_	k_q_ [M^−1^·s^−1^]
BSA	5.41	0.972	8.58 × 10^12^
BSA-Na[B(Ph)_4_] (1:3)	3.12	0.965	4.94 × 10^12^
BSA-Na[B(Ph)_4_] (1:7)	1.98	0.987	3.15 × 10^12^
BSA-Na[B(Ph)_4_] (1:15)	1.35	0.962	2.14 × 10^12^

**Table 3 molecules-26-06565-t003:** The secondary structure content (%) of BSA at different Na[B(Ph)_4_]:BSA molar ratios in 10 mM Caco buffer at pH 7 in the temperature range 298.15–368.15 K, revealed from CD measurements.

T [K]	BSA	Na[B(Ph)_4_]:BSA	Na[B(Ph)_4_]:BSA
		2.5:1	7:1
		(Molar Ratio)	(Molar Ratio)
	The Percentage Content of α-Helix (H); β-Structure (S); β-Turn (Trn); Random Coil (Unrd) [%]		
298.15	59.8 (H); 6.7 (S)	62.5 (H); 5.2 (S)	59.2 (H); 6.1 (S)
	11.1 (Trn); 22.3 (Unrd)	10.5 (Trn); 21.1 (Unrd)	11.5 (Trn); 23.1 (Unrd)
308.15	60.5 (H); 8.0 (S)	63.9 (H); 4.5 (S)	59.5 (H); 6.8 (S)
	11.7 (Trn); 20.3 (Unrd)	10.3 (Trn); 21.3 (Unrd)	11.5 (Trn); 22.6 (Unrd)
318.15	56.0 (H); 7.3 (S)	61.3 (H); 5.8 (S)	58.2 (H); 7.8 (S)
	13.4 (Trn); 23.6 (Unrd)	10.9 (Trn); 22.6 (Unrd)	12.1 (Trn); 22.4 (Unrd)
328.15	50.8 (H); 10.5 (S)	58.0 (H); 7.0 (S)	56.0 (H); 8.4 (S)
	14.1 (Trn); 24.9 (Unrd)	12.5 (Trn); 22.6 (Unrd)	12.1 (Trn); 23.8 (Unrd)
338.15	44.4 (H); 11.5 (S)	47.7 (H); 11.3 (S)	50.2 (H); 8.4 (S)
	16.7 (Trn); 27.5 (Unrd)	15.6 (Trn); 25.9 (Unrd)	15.8 (Trn); 25.9 (Unrd)
348.15	36.4 (H); 15.0 (S)	39.1 (H); 14.4 (S)	50.5 (H); 9.1 (S)
	19.8 (Trn); 29.4 (Unrd)	18.6 (Trn); 28.3 (Unrd)	15.7 (Trn); 25.5 (Unrd)
358.15	25.4 (H); 21.2 (S)	35.4 (H); 15.6 (S)	23.7 (H); 22.4 (S)
	24.1 (Trn); 29.8 (Unrd)	20.7 (Trn); 27.7 (Unrd)	22.9 (Trn); 31.0 (Unrd)
368.15	25.3 (H); 22.2 (S)	22.0 (H); 24.9 (S)	22.6 (H); 22.5 (S)
	22.6 (Trn); 30.4 (Unrd)	22.3 (Trn); 31.0 (Unrd)	22.5 (Trn); 32.1 (Unrd)

**Table 4 molecules-26-06565-t004:** LIE free-energy decomposition values for [B(Ph)_4_]^−^ ligands in the [B(Ph)_4_]^−^–BSA complex. Three [B(Ph)_4_]^−^ ions (no 2, 6 and 11) did not form stable complexes with BSA throughout the simulation. ΔG_ele_, ΔG_vdW_ and ΔG_tot_ correspond to the electrostatic component, van der Waals component and total energy, respectively.

No of [B(Ph)_4_]^−^	ΔG_ele_, [kcal mol^−1^]	ΔG_vdW_, [kcal mol^−1^]	ΔG_tot_, [kcal mol^−1^]
1	−0.3 ± 0.1	−21.0 ± 2.5	−21.3 ± 2.6
2	N/A	N/A	N/A
3	−0.2 ± 0.1	−23.0 ± 3.2	−23.3 ± 3.2
4	−0.6 ± 0.2	−22.5 ± 2.9	−23.1 ± 2.9
5	−0.2 ± 0.1	−23.4 ± 3.4	−23.6 ± 3.4
6	N/A	N/A	N/A
7	−0.4 ± 0.2	−18.6 ± 5.7	−18.7 ± 5.9
** 8 **	**−0.6 ± 0.1**	**−29.9 ± 2.5**	** −30.5 ± 2.5 **
9	−0.3 ± 0.1	−15.8 ± 2.5	−16.1 ± 2.5
** 10 **	**0.7 ± 0.1**	**−32.2 ± 2.6**	** −32.8 ± 2.6 **
11	N/A	N/A	N/A
12	−0.3 ± 0.2	−21.2 ± 2.9	−21.5 ± 2.9
13	−0.2 ± 0.1	−15.2 ± 1.9	−15.4 ± 1.9
**14**	**−0.6 ± 0.2**	**−28.5 ± 3.2**	**−29.0 ± 3.3**
15	−0.3 ± 0.2	−20.8 ± 3.8	−21.1 ± 3.9

## Data Availability

The data presented in this study are available on request from the corresponding author.

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
