# Peer review of "Effect of Tetraphenylborate on Physicochemical Properties of Bovine Serum Albumin"

_molecules, 2021, doi:10.3390/molecules26216565_

Round 1

Reviewer 1 Report

Grabowska et al described an experimental study on the interactions between tetraphenylborate and bovine serum albumin. They discovered two sets of structurally distinctive binding sites in BSA and the increase of thermal stability of the alfa-helical content. This is an interesting study, however, some critical flaws should be addressed.

  1. It is not convincing that the binding sites reported in the paper.  Do the binding sites form hydrophobic cavities?  Do any residue substitutions at the binding sites inhibit the interactions?
  2. How do the binding conformation in Fig 11 obtained? More detailed information should be provided. 
  3. The two binding regions should be compared to illustrate their similarities and differences.
  4. How can the current conclusions be useful to other types of compounds of a hydrophobic nature?  Are the binding  not specific?

Reviewer 2 Report

The manuscript “Effect of Tetraphenylborate on Physicochemical Properties of Bovine Serum Albumin” by Wyrzykowski and co-workers presents an integrative experimental/computational study on the binding of tetraphenylborate with BSA. Several instrumental techniques were used to define the binding constants and the number of binding sites, namely Isothermal Titration Calorimetry (ITC) and Steady-state fluorescence spectroscopy. Furthermore, the effect of the binding on the protein structure has been assessed by Differential Scanning Calorimetry (DSC) and Circular Dichroism (CD). Finally, a Molecular Dynamics (MD) study was used to characterize the identified binding sites at molecular level. All the data of this work was discussed on top of a previous published study by the same authors on the interaction of sodium dodecyl sulfate (SDS) with BSA and new competition experiments have been performed.

As a result the authors unveiled two main binding sites in the regions containing Trp134 (Site I) and Trp213 (Site II). On one hand a similar behavior between SDS and tetraphenylborate is characterized having both compounds close binding constant toward the first specific site I. The second site II become populated only after the saturation of the specific site. On the other hand, the effect on the protein structure upon binding shows differences between the two compounds. While the binding of tetraphenylborate to Site I does not significantly affect the whole conformation of BSA, SDS induces appreciable confrontational transitions. MD analysis shows a picture of the putative binding sites at the molecular level.

The manuscript is generally well written, the introduction provide a good background, the methods are clearly presented and the study is technically correct. The results could be of interest for the readers, are adequate for the scope of the journal and particularly centered on the focus of the special issue. Despite these general good considerations, the manuscript lacks in proper discussion and investigation of the key factors governing the specificity of the binding. Moreover, the introduction should be improved with several recent references and more specific information should be given about the interest of this system in the context of drug-design. For these reasons this referee suggest a reconsideration after major revisions (see the following points).

1) Albumins are one of the main objects of study in the field of bioinorganics and metallo-drug design. So, this aspect should be improved in the introduction and several recent article should be cited. For example: Chem. Sci. 2016, 7, 6635-6648; Chem. Eur. J. 2020, 26, 11316-11326; Inorg. Chem. Front., 2021, 8, 1951-1974

2) Introduction should clearly give a clear picture of the of the general interest of this study with a paragraph focused in the aims and the main message of the paper. What are really the recent insights into potential-drugs interactions with albumins? What is new and timely? At the end of the intro, it is also not clear what is the main message and relevant points of the paper that should be emphasize at this stage.

3) The key factors governing the specificity of the binding should be deeply discussed, particularly the molecular interaction of site I compared with those of Site II that appear to be less specific. In this context a discussion of the binding free energy toward Site I and Site II is lacking. For example which of the 15 poses reported in table 4 are relative to Site I and Site II. There are DG differences? What is the origin of these differences? This kind of analysis should push forward the knowledge of drug-design.

4) The criteria used to define the total time of MDs (convergence criteria) should be described in the methods, as well as the criteria followed to choose the frames of the trajectory used for binding free energy calculations.

Reviewer 3 Report

The manuscript by Grabowska et al reports on interaction of tetraphenylborate ions with bovine serum albumin (BSA). Such interactions can influence the protein properties, thus introducing bias in cytotoxicity assays. The work is mostly done using a combination of various experimental methods, with some addition of molecular modeling to explain experimental results. Authors conclude that multiple binding sites in BSA structure can be involved in the tetraphenylborate ion binding with different effect on the protein properties.

The study overall is interesting. As a computational biologist, I will leave the experimental part to specialists in the corresponding fields, and limit my criticism to the molecular modeling part.

1. Text and logic of the manuscript, Introduction and Conclusions in particular, are sometimes hard to follow and should be revised for clarity.

2. What is the value of the molecular modeling part? It identified novel binding sites, or confirmed binding of new ions to previously established sites? This key point has to be clarified.

3. The molecular modeling part seems to investigate protein-ion interactions by the means of classical/conventional molecular dynamics simulation using the AMBER package. The choice of methodology with respect to the task was not clear to me. I would rather use molecular docking to attempt ion binding to all potential sites in a protein structure, followed by selection of the confirmed interacting partners. Instead, authors simulate free binding of ion molecules from the solvent to the protein, studied by independent multiple runs to collect more data. Such an approach indeed was previously used in the literature, but to investigate the binding mechanisms in dynamics, rather than to confirm binding sites and estimate binding energies. The rationale for selection of this particular computational methodology was not clearly stated, and should be provided during revision.

4. Although I dont have personal experience with the ion in question, the binding energy estimate by molecular modeling of -30 kcal/mol seems too large to me for this particular ligand. Can such number be backed experimentally? Or by alternative computational methods?

5. I find the following statement very strange: "Three [B(Ph)4]- ions (no 2, 6 and 11) were unstable throughout the simulation". What exactly happened to the "unstable" ions?! As I understood, this was a classical MD simulation, i.e. the covalent bonds in molecule topologies were hard-fixed. Then, how did the ions become unstable?

Round 2

Reviewer 1 Report

The authors have addressed some of my concerns. However, I did not agree with some of the answers.

  1. The evidence for ligand binding sites is indirect. There should be some direct evidence to get the conclusion, such as residue substitutions to block the binding.
  2. It is not true that“Docking small ligands usually allows for finding a single binding site and not the multiple ones in case when one binding site is essentially more preferable than other ones.” The authors can refer to some blind protein-ligand docking methods, which usually result in multiple binding sites.
  3. In most cases, MD is not used for predicting binding sites. The nowadays MD simulation which usually modeling the conformation changes in tens of nano seconds, cannot intensively explore the surface of a protein which needs order of magnitude larger time consumption. Thus although MD is a well-established method in molecular simulation, it is not yet strictly benchmarked that MD can result in better prediction in discovering binding sites/modes than molecular docking.  So the answer to point 2 is not convincing.
  4. “POINT 3. The two binding regions should be compared to illustrate their similarities and differences.” Here the similarities and differences indicate the binding modes and residue composition.

Reviewer 2 Report

In the revised version of the manuscript the authors addressed all the comments of this referee, however not all the answers and the new parts of the revised manuscript are fully satisfactory. Therefore, this referee suggests again major revisions before consideration for publication. See the following points:

3) Round 1: The key factors governing the specificity of the binding should be deeply discussed, particularly the molecular interaction of site I compared with those of Site II that appear to be less specific. In this context a discussion of the binding free energy toward Site I and Site II is lacking. For example which of the 15 poses reported in table 4 are relative to Site I and Site II. There are DG differences? What is the origin of these differences? This kind of analysis should push forward the knowledge of drug-design.

ANSWER.To address this point of the Reviewer, Table 4 was extended: electrostatic and vdW components are now explicitly provided in the revised manuscript. Accordingly, the discussion on the differrences between sites I and II is added to the section 2.5: “According to LIE calculations, site I has slightly higher affinity towards [B(Ph)4]-than site II due to the more favourable van der Waals component of the free binding energy.”

Round 2: This referee regrets to stress again on this point, a single sentence stating that the higher affinity is due to vdW interaction is not enough to give the possibility to other researchers to take advantage of this work in the future. A clear description of the specific interactions (which aa interact with which substituent of the molecule? which kind of vdW interaction, pi-stacking, cation-pi, pi-pi?) including a deep (as quantitative as possible) discussion about the differences between Site I and II should be reported.

4) Round 1: The criteria used to define the total time of MDs (convergence criteria) should be described in the methods, as well as the criteria followed to choose the frames of the trajectory used for binding free energy calculations.

ANSWER.This point is addressed in the revised manuscript in section 3.6.2: “Such length of the MD simulation was shown to be sufficient for a system of ions with a protein, which was substantially bigger than BSA, allowing for proper prediction of the ion binding sites [41].” and in section 3.6.3: “Frames from the MD simulation were carefully examined with VMD [45]. The last 40% of the simulation was chosen for the analysis since in the corresponding frames no more events of ions dissociation/association were observed.” New references were added, correspondingly (see the revised version of the manuscript).

Round 2. This referee understands the reasons of the authors, but again the answer is not satisfactory since the MD time until convergence cannot be derived from a previous published paper on a different system. In fact, independently from the dimension of the protein, the potential flexibility of the protein as well as the mobility of the ligands and the induced flexibility are intrinsic characteristics of the specific system and must be evaluated to ensure the convergence of the MD. The authors could refer to a recent paper (Front. Chem. 7:211, doi: 10.3389/fchem.2019.0021) in which these aspect are well explained and several techniques proposed. This referee ask to the authors to perform such kind of analysis to ensure MD convergence and reproducibility of their calculations.

Also the statement that “Frames from the MD simulation were carefully examined with VMD. The last 40% of the simulation was chosen for the analysis since in the corresponding frames no more events of ions dissociation/association were observed.” is not enough to ensure the reproducibility of the calculations reported in this work. Clear mathematical criteria must be used to ensure the stabilization of the ligands inside the binding sites. Such an example RMSD along the trajectory could be used. This referee kindly invite the authors to perform such analysis reporting the details as information for the community.

Reviewer 3 Report

Authors have responded to my criticism and revised the manuscript as suggested. The manuscript can be published in the present form.

Author Response

We would like to express our gratitude to the Reviewer for the useful comments and valuable remarks.